

# Two pathways of how SST anomalies drive the interannual variability of autumnal haze days in the Beijing–Tianjin–Hebei region, China

**Jing Wang[1], Zhiwei Zhu[1*], Li Qi[1], Qiaohua Zhao[1], Jinhai He[1], and Julian X. L. Wang[2]**

[1]Key Laboratory of Meteorological Disaster, Ministry of Education (KLME)/Joint International Research Laboratory of Climate and Environment Change (ILCEC)/Collaborative Innovation Center on Forecast and Evaluation of Meteorological Disasters (CIC-FEMD), Nanjing University of Information Science and Technology, Nanjing, China

[2]Air Resources Laboratory, National Oceanic and Atmospheric Administration, College Park, MD, USA

[*] *Correspondence to:* Zhiwei Zhu (zwz@nuist.edu.cn)

**Abstract.** Analogous to the circumstances in wintertime, the increasing severity of autumnal haze pollution over the Beijing–Tianjin–Hebei (BTH) region may also lead to impairment of the socioeconomic development and human health in this region. Despite manmade aerosol emissions, the interannual variability of autumnal (September–October–November) haze days (AHD) in the BTH region ($AHD_{BTH}$) is apparently tied to the global and regional meteorological anomalies. The present study suggests that an above-normal $AHD_{BTH}$ is closely associated with the simultaneous sea surface temperature (SST) warming in two regions [over the North Atlantic subtropical sector (R1) and over the western North Pacific sector (R2)]. When the autumnal SST warming in R1 and R2 are both remarkably significant, the joint impacts can greatly enhance the likelihood of a higher $AHD_{BTH}$. Observational and simulation evidence suggests that SST anomalies can affect the variation in $AHD_{BTH}$ via two different pathways. Firstly, SST warming in R1 can induce a downstream mid-latitudinal Rossby wave train, leading to a barotropic high-pressure and subsidence anomaly over the BTH region. Secondly, SST warming in R2 can also result in air subsidence over the BTH region through an anomalous local meridional cell. Through these two distinct pathways, localized meteorological circumstances conducive to a higher $AHD_{BTH}$ (i.e., repressed planetary boundary layer, weak southerly airflow, and warm and moist conditions) can be established.

## 1 Introduction

Aerosol particles (APs) are ubiquitous in the ambient air. Through aerosol-induced forcing, APs can exert profound impacts on regional and large-scale circulation (e.g., Chung et al., 2002; Lau and Kim, 2006; Lau et al., 2006; Liu et al., 2009; Li et al., 2016; Wu et al., 2016), as well as global warming (e.g., Charlson et al., 1992; Tett et al., 1999; Zhang et al., 2016). Notably, due to the property of light extinction related to high concentrations of APs, especially fine particulate matter [i.e., particulate matter (PM) with an aerodynamic diameter of 2.5 μm or less ($PM_{2.5}$)] (Guo et al., 2014; Wang et al., 2014; Li et al., 2017; Seo et al., 2017; Chen et al., 2018; Luan et al., 2018), severe haze weather with low visibility and high concentrations of gas pollutants can readily occur (Chen et al., 2012; Li et al., 2016; Ding et al., 2017; Seo et al., 2017; Chen et al., 2018).

In recent decades, observational evidence suggests that China has become one of the most severe AP-loading regions in the world (Tao et al., 2016; Li et al., 2016), arguably because of the country's rapid industrialization and urbanization (Xu et al., 2015; Zhang et al., 2016). High concentrations of APs can lead to the formation of severe haze weather via complicated interactions (Wang et al., 2014). Haze weather is not only harmful to the human respiratory and cardiovascular systems (Pope III and Dockery, 2006; Tie et al., 2009; Chen et al., 2013; Xu et al., 2013), but also influences vehicular traffic and crop yields (Chameides et al., 1999; Wu et al., 2005). As a result, haze pollution has received considerable attention from both the government and the public. Unfortunately, on the one hand, overwhelming industrialization leads to more severe haze contamination over the Beijing–Tianjin–Hebei (BTH) region (Yin et al., 2015); whilst



on the other hand, the trumpet-shaped topography (Fig. 1) of the region is unfavorable for the dissipation of air pollution, thus making the BTH region home to some of the worst haze weather in China. Since the BTH region is the most economically developed region in North China and is at the heart of Chinese politics and culture (not least because it is home to the capital city, Beijing, and Xiongan New Area, for instance), severe haze pollution in this region has become a critical issue (e.g., Mu and Zhang, 2014; Yin et al., 2015; Wang, 2018), especially since the unprecedented severe haze event in North China in January 2013 (Wang et al., 2014; Zhang et al., 2014; Mu and Zhang, 2014; Tao et al., 2014; Zhang et al., 2015).

To date, numerous efforts have been made to explore the causes of wintertime haze pollution over the BTH region and its surroundings, and these efforts roughly fall into three categories of results from the climatological perspective. The first category features studies that have reported that the joint effects of the emissions of various sources of APs (e.g., Cao et al., 2007; Guo et al., 2011; Zhu et al., 2016) and climate anomalies (e.g., Chen and Wang, 2015; Wang and Chen, 2016; Yin and Wang, 2016a; Cai et al., 2017; Yin et al., 2017; Yin and Wang, 2018; Wang, 2018) may have brought about the increasing severity of haze pollution over China in recent decades. The second category of studies, meanwhile, underlines the causality of the variation in winter haze days in eastern and northern China from the perspective of climate anomalies (e.g., Li et al., 2016; Yin and Wang, 2016b; Yin and Wang, 2018; Pei et al., 2018). For instance, a weakened East Asian winter monsoon (EAWM) system has been suggested as being responsible for above-normal numbers of winter haze days (e.g., Niu et al., 2010; Li et al., 2016; Yin and Wang, 2016a; Yin and Wang, 2017; Yin et al., 2017); plus, the EAWM's variability has been shown to be significantly tied to the East Atlantic–West Russia pattern (Yin et al., 2017; Yin and Wang, 2017) and Eurasian pattern (Zhang et al., 2016; Yin et al., 2017). The third category of studies focuses on the external forcings associated with the variability of winter haze days. These forcings include the sea surface temperature (SST) (e.g., Gao and Li, 2015; Wang et al., 2015; Yin and Wang, 2016a; Yin et al., 2017), Arctic sea ice (e.g., Wang et al., 2015; Zou et al., 2017), Eurasian snowpack (e.g., Yin and Wang, 2017; Yin and Wang, 2018), and the thermal conditions on the Tibetan Plateau (e.g., Xu et al., 2016). However, most of these previous works have focused on wintertime, with little attention having been paid to other seasons.

Autumn is a transitional season from the wet and hot conditions of summer to the dry and cold conditions of winter. The weather in autumn over the BTH region is climatologically quite pleasant, with favorable temperatures and light winds. Outdoor activities and tourism are therefore important, economically, in the autumn season. However, notably, autumn is also a season in which haze weather frequently occurs in the BTH region (Chen and Wang, 2015), and the number of autumnal haze days (AHDs) has increased remarkably in recent years. Such an increase in the number of haze days is a potential threat to the outdoor activities and tourism that, as mentioned, are so important to the region at this time of year. Therefore, research into the causes of the interannual variation in AHDs in the BTH region ($AHD_{BTH}$) is imperative. Such work not only provides scientific support to the year-to-year scheduling of anthropogenic emissions for dealing with autumnal haze pollution, but also helps the government with facilitating the arrangement of tourism and outdoor activities. However, as already mentioned, compared to the myriad publications on wintertime haze pollution, autumn haze pollution over the BTH region has attracted far less attention, with only a few case studies on atmospheric circulation having been reported (Yang et al., 2015; Gao and Chen, 2017; Wang et al., 2018). It was this knowledge gap that motivated us to revisit the variability of $AHD_{BTH}$. Considering that the SST acts as a crucial driver of large–scale climate variability (e.g., Wang et al., 2009; Zhu et al., 2014; He and Zhu, 2015; Xiao et al., 2015; Zhu and Li, 2017; Zhu, 2018), we aimed to figure out the underlying air–sea interaction mechanisms for the interannual $AHD_{BTH}$ variability in the present study.



The remainder of this paper is organized as follows. Section 2 introduces the data, model and
methodology. Section 3 presents the atmospheric anomalies associated with $AHD_{BTH}$. Section 4
addresses the mechanisms and pathways of SST anomalies (SSTAs) in driving the interannual
variations of $AHD_{BTH}$. Conclusions and further discussion are provided in the final section.
**2 Data, model and methodology**
**2.1 Data**
The data used in this study are as follows: (1) monthly mean planetary boundary layer height
(PBLH), with a 1° × 1° horizontal resolution, from the European Centre for Medium-Range
Weather Forecasts Interim Reanalysis (ERA-Interim) (Dee et al., 2011); (2) monthly mean
atmospheric data, with a 2.5° × 2.5° horizontal resolution, from the National Centers for
Environmental Prediction (NCEP)–National Center for Atmospheric Research (NCAR)
Reanalysis I (NCEP/NCAR) (Kalnay et al., 1996); and total cloud cover (entire atmosphere
considered as a single layer; 192 × 94 points in the horizontal direction), also from NCEP/NCAR;
(3) monthly mean SST, with a 2° × 2° horizontal resolution, of the Extended Reconstructed SST
dataset, version 5 (ERSST.v5; Huang et al., 2017), from the National Oceanic and Atmospheric
Administration (NOAA); (4) global monthly precipitation data, with a 2.5° × 2.5° horizontal
resolution, from NOAA's precipitation reconstruction (Chen et al., 2002); (5) ground-timing
observation datasets, at 02:00, 08:00, 14:00 and 20:00 BLT (Beijing local time), from the National
Meteorological Information Center of China. The temporal coverage of the PBLH data is from
1979 to 2017, while the remaining datasets are from 1960 to 2017. Here, boreal autumn refers to
the seasonal mean for September–October–November (SON).
**2.2 Model**
The numerical model used here is an anomaly atmospheric general circulation model (AGCM)
based on the Geophysical Fluid Dynamics Laboratory (GFDL) global spectrum dry AGCM (Held
and Suarez, 1994), which is employed to investigate the mechanisms for the atmospheric
responses to the specified SST-induced heating. The horizontal resolution is T42, with five evenly
spaced sigma levels ($\sigma = p/ps$; interval: 0.2; top level: $\sigma = 0$; bottom level: $\sigma = 1$). A realistic
autumn mean state, obtained from the long-term mean of the NCEP/NCAR reanalysis data, is
prescribed as the model basic state. This model has been used to unravel the eddy–mean
interaction over East Asia and its downstream impacts on North American climate (Zhu and Li,
130 2016, 2018).

**2.3 Methodology**
The definition of a haze day in the present study is identical to that used in previous studies (Chen
and Wang, 2015; Yin et al., 2017; Pei et al., 2018). It is based on the ground-timing observations
of relative humidity, visibility and wind speed. It is important to point out that the visibility
observations switched from manual to automatic in 2014, and the visibility threshold for haze was
thus also slightly modified from then on. However, the continuity of the data was not affected.
Following Zhang et al. (2016), the mean number of haze days ($\overline{NHD}$) for $AHD_{BTH}$ was computed
by:





$$\overline{\text{NHD}}=\frac{1}{n}\sum_{i=1}^{n}N \qquad (1)$$
where $n$ (here, $n = 20$) is the number of meteorological sites distributed within the BTH region
(Fig. 1), and $N$ denotes the number of haze days at a site for each autumn.
Similar to the approach proposed by Zhu and Li (2017), the 9-yr running mean of the $AHD_{BTH}$
was used to represent the interdecadal component of the $AHD_{BTH}$, whereas the interannual
component was obtained by removing the interdecadal component from the raw $AHD_{BTH}$. Since
there is a tapering problem when calculating the running mean, the first four years and the last
four years of the interdecadal component of the $AHD_{BTH}$ could be estimated by the mean value of
the available data with a shorter window. For example, the interdecadal component of the $AHD_{BTH}$
for 2016 and 2017 could be obtained by the mean of 2012–17 and 2013–17, respectively. Note
that the temporal correlation coefficients (TCCs) between the $AHD_{BTH}$ and every single site were
all positive and significant (Fig. 1), indicating coherency in the interannual variability of haze days
over the BTH region; plus, the distribution of these sites was also fairly even. Therefore, the
interannual component of the $AHD_{BTH}$ could be used as a good representation of the year-to-year
pollution state over the whole BTH region in autumn.
Linear regression, composite analysis and correlation were used to examine the associated
circulation and SSTAs. The two-tailed Student's $t$-test was employed to evaluate the statistical
significance of these analyses. The wave activity flux (WAF; Takaya and Nakamura, 2001) was
calculated to depict the tendency of Rossby wave energy propagation.
**3 Atmospheric anomalies associated with the interannual changes of $AHD_{BTH}$**
Figure 2 illustrates the time series of the raw $AHD_{BTH}$, along with its interdecadal and interannual
components. A prominent feature is that the $AHD_{BTH}$ displays both interannual and interdecadal
variability. On the interdecadal timescale, the $AHD_{BTH}$ was below average during 1960–1975 and
the late-2000s, but above average during 1975–2003, and it increased dramatically after 2009. On
the interannual timescale, the $AHD_{BTH}$ presents large differences year on year. For example, the
$AHD_{BTH}$ was at its lowest in 2012, but peaked in 2014. Since the interannual variability explains
most of the variances in the $AHD_{BTH}$ variability, in this study we only investigate the atmospheric
anomalies and unravel the underlying physical mechanisms and pathways associated with the
$AHD_{BTH}$ on the interannual timescale.
Close scrutiny of the large-scale and localized dynamic and thermodynamic fields associated with
the $AHD_{BTH}$ should help in advancing our understanding of the possible underlying mechanisms.
In this regard, we firstly examine the climatological mean autumnal 500-hPa geopotential height
(Z500), 850-hPa winds (UV850) and total cloud, along with the surface relative humidity and
surface air temperature that potentially impact the climate over the BTH region (Fig. 3). There is a
shallow mid-tropospheric trough over coastal East Asia (Fig. 3a), which resembles the trough in
winter (Zhao et al., 2018; Pei et al., 2018) but with a smaller magnitude. Behind the trough, a clear
anticyclonic circulation appears over the central-eastern China, with remarkable
westerly/northwesterly winds dominating the BTH region (Fig. 3a). Cold and dry air from higher
latitudes is advected by the winds, and the BTH region is thus much cooler and drier and has less
cloud than other regions at the same latitudes (e.g., the central portion of Japan). As such, the
autumnal BTH region features breezy and windy conditions climatologically, with low surface
relative humidity (Fig. 3b), reducing the likelihood of haze there via the effect of cold
advection/ventilation. Note, however, that if the breezy conditions are interrupted, haze pollution



may be enhanced. One may ask whether a higher $AHD_{BTH}$ is related to the interference of such
breezy conditions. Figures 4 and 5 were therefore plotted to examine the associated atmospheric
parameters/circulations. For simplicity, the regression and composite analyses in this study
reported hereafter are interpreted with respect to positive phase of $AHD_{BTH}$ anomalies only.
Previous studies have revealed that haze pollution is closely correlated with local meteorological
parameters in the planetary boundary layer (e.g., You et al., 2017; Chen et al., 2018). Figure 4
suggests that an above-normal $AHD_{BTH}$ is tied to a localized enhancement of surface relative
humidity (Fig. 4a) and temperature (Fig. 4b), along with suppressed surface wind speed (Fig. 4c),
sea-level pressure (SLP) (Fig. 4d) and PBLH (Fig. 4e). Specifically, it seems that autumnal haze
pollution is more significantly correlated with temperature and PBLH. So, what causes the above
anomalous parameters that are favorable for a higher $AHD_{BTH}$?
Figure 5 shows the associated large-scale atmospheric circulation anomalies at different levels of
troposphere. From Figs. 5a–5d, the most noticeable feature is that there is a planetary-scale,
quasi-barotropic Rossby wave train emanating from the North Atlantic subtropical sector. In
addition to an anticyclonic anomaly centered over the North Atlantic subtropics, this
teleconnection pattern has another two pairs of anomalous cyclones (low pressure) and
anticyclones (high pressure) stretching across Eurasia to the North Pacific, i.e., a cyclonic
anomaly centered over the ocean south of Greenland, an anticyclonic anomaly centered over
Scandinavia, a cyclonic anomaly centered over the adjacent central Siberia, and a Northeast Asian
anticyclonic anomaly centered over the Sea of Japan (SJ). In general, based on the regressed
atmospheric fields, the teleconnection has a much larger amplitude in the upper troposphere (Fig.
5a), rather than in the mid-troposphere (Fig. 5b) and lower troposphere (Fig. 5c).
Among all the height anomalies within the teleconnection, the anomalous quasi-barotropic
Northeast Asian anticyclonic anomaly centered over the SJ ($A_{SJ}$) plays a direct role in driving a
higher $AHD_{BTH}$. The related physico-meteorological causes are as follows: There are
southerly/southeasterly anomalies along the western flank of the $A_{SJ}$ in the lower troposphere (Figs.
5c and 5d), manifesting the capability of suppressed atmospheric horizontal diffusion and thus
favoring a buildup of substantial local and nonlocal APs and warmer moisture over the BTH
region (Yang et al., 2015; Yang et al., 2016) under the specific topographical forcing of the
Taihang Mountains and Yan Mountains (Fig. 1). On the other hand, the significant positive
pressure anomaly in the mid-to-upper parts of the $A_{SJ}$ (Figs. 5a and 5b) not only impedes the
intrusion of cold air into the BTH region, but also facilitates consistent air subsidence over the
BTH region and its surrounding areas (Fig. 4f), resulting in the decrease of the PBLH and
amplification of static stability (i.e., the dampened vertical dispersion of the atmosphere).
Consequently, the meteorological conditions connected to a higher $AHD_{BTH}$ are quite different
from the climatological characteristics (Fig. 3).
To summarize, the $A_{SJ}$ and the associated subsidence can induce the capacity for suppressed local
horizontal and vertical dispersion over the BTH region and its surrounding areas, as shown in the
above-mentioned anomalous parameters in the boundary layer (Fig. 4); and these parameters are
further responsible for the accumulation and secondary formation/hygroscopic growth of APs
(Jacob and Winner, 2009; Ding and Liu, 2014; Mu and Liao, 2014; Jia et al., 2015). As such, the
haze pollution over the BTH region is readily established within a narrow space. The question of
how the above-normal $AHD_{BTH}$ is stimulated could plausibly be transferred into questioning the
pathways of how the $A_{SJ}$ is developed and sustained. In fact, the $A_{SJ}$ and the associated air
subsidence are modulated by SSTAs. We tackle this issue in the next section.



## 4 Possible mechanisms and pathways

### 4.1 Observational diagnoses

Figure 5c shows that an above-normal $AHD_{BTH}$ is closely correlated with SST warming in two key regions: the North Atlantic subtropical sector (R1: 22°–32°N, 90°–40°W), and the western North Pacific sector (R2: 10°–30°N, 108°–140°E). Meanwhile, from Fig. 5e we can discern that enhanced and significant precipitation appears to the north of R1, indicating an active atmospheric response to the SST warming over R1; whereas, there is an insignificant positive precipitation signal over R2 and its surrounding areas. Figure 6 further depicts that the SON SSTs over both R1 and R2 are positively correlated with $AHD_{BTH}$, and the TCC between the $AHD_{BTH}$ and SON SST over R1 (R2) is 0.45 (0.28), exceeding the 99% (95%) confidence level. By virtue of the above analyses, we speculate that the SST over R1 may play a more important role than that over R2 in driving a higher $AHD_{BTH}$. Note, however, that when the SON SSTs over R1 and R2 are both obviously elevated, the $AHD_{BTH}$ is more likely to be higher than normal, such as in 1980, 1987 and 2015. Furthermore, as indicated above, the $AHD_{BTH}$ is closely correlated with the $A_{SJ}$ and the associated air subsidence, which allows us to speculate that the positive SSTAs over R1 and R2 might drive the interannual variability of $AHD_{BTH}$ by modulating the intensity of the $A_{SJ}$ and associated subsidence. To validate this hypothesis, we firstly examine pathway of SSTAs over R1 in driving $AHD_{BTH}$.

Figure 5c suggests that the SST warming in R1 may induce larger-area concomitant low-level easterly anomalies, which mainly form over the southeastern portion of R1 and the area to its south. In such a scenario, an anticyclonic anomaly is induced (Fig. 5c), with its center to the northeast of R1. Along the western flank of this anticyclonic anomaly, warm and moist airflows move northwards. When these warm and moist airflows meet cold air mass in the areas to the north of R1, enhanced precipitation is thus generated (Fig. 5e). Meanwhile, the resultant enhanced rainfall condensation heating induces a cyclonic anomaly to its north, thereby exciting the other two pairs of the aforementioned teleconnection pattern along the westerly jet, as demonstrated by the Rossby wave train induced by SST warming in R1 (Figs. 7 and 8). Specifically, from the regressed SON UV850 (Fig. 7), we can see that the SST warming in R1 can indeed induce a significant low-level teleconnection pattern arising from the North Atlantic subtropics, bearing a close resemblance to that in Fig. 5c; and to the north of R1, where the rainfall condensation heating is triggered, the corresponding WAF exhibits a distinctive arc-shaped trajectory, perturbing the other two pairs of cyclones and anticyclones of the teleconnection (Fig. 8). This teleconnection extends from the North Atlantic towards Scandinavia, goes through the Eurasia and arrives at the western North Pacific. Therefore, by means of this trajectory, Rossby wave energy in the middle (Fig. 8b) and upper (Fig. 8a) troposphere may propagate southeastwards into the $A_{SJ}$ and its surrounding region, favoring the formation/sustainability of the $A_{SJ}$ and the associated air subsidence. In this context, the associated meteorological parameters (Fig. S1), which resemble those tied to a higher $AHD_{BTH}$ (Fig. 4), might increase the likelihood of SON haze pollution over the BTH region. Again, this induced teleconnection is quasi-barotropic in structure, with its magnitude larger in the upper troposphere (Fig. 8a), which is consistent with that in Fig. 5a.

When focusing on region R2 (Fig. 9a), we find that, corresponding to the SSTAs over R2, there exists a cyclonic anomaly to the west of R2. Besides, substantial SSTA-induced low-level easterly anomalies are mainly located to the southeast of R2; plus, a huge anticyclonic anomaly to the northeast is excited, with its center situated over the northern Pacific. In such a scenario, R2 is thoroughly penetrated by significant warm and humid airflows transported from the eastern flank



of the cyclonic and the western flank of anticyclonic anomaly respectively (Fig. 9a), warming the
SST over R2. Furthermore, the airflow convergence primarily occurs over the southwestern
portion of R2, where the strongly significant and positive rainfall anomaly is triggered (Fig. 9b).
Thus, the enhanced significant rainfall heating perturbation may greatly intensify the ascending
motion over R2 and the adjacent region, resulting in subsidence over the BTH region and
Northeast Asia via an anomalous local meridional cell (Fig. 10a). As such, the BTH region and its
adjacent areas are dominated by significant warm temperatures in the middle and upper
troposphere (Fig. 10b), leading to the maintenance and reinforcement of the $A_{SJ}$ and the downward
motions over the BTH region, as well as the regional low-level stability. Under such
circumstances, the vertical transport of APs is restricted (Zhang et al., 2014; Pei et al., 2018), and
the near-surface winds are weakened (Li et al., 2016). Meanwhile, the parameters associated with
SST warming in R2 (Fig. S2) also support the formation of haze weather over the BTH region.
**4.2 Numerical model simulations**
Two experiments were conducted to further validate the above-mentioned two pathways in how
SSTAs drive the variation of $AHD_{BTH}$. The first experiment (H_NAS) simulated the responses to
the heating induced by SSTAs over R1 (Fig. 11). H_NAS was imposed with a specified heating
centered over the region to the north of R1 (center: 37.67°N, 64.69°W) that largely matched with
the SON positive rainfall anomaly as shown in Fig. 5e. The second experiment (H_WNP)
mimicked the responses to the prescribed heating over the neighboring areas of R2 (center:
15.35°N, 109.69°E; Fig. 12), where the corresponding regressed precipitation rate was the most
significant and amplified, as exhibited in Fig. 9b. The heating had a cosine-squared profile in an
elliptical region in the horizontal direction. The maximum heating, with 1 K day$^{-1}$ amplitude, was
set to be at 300 hPa.
Figure 11 presents the 200- and 500-hPa geopotential height and wind responses to the specified
heating over the North Atlantic subtropical region. As anticipated, the equilibrium state (mean
output from day 40 to day 60) of the Z200 (Fig. 11a) and Z500 (Fig. 11b) responses to the heating
resembles the aforementioned teleconnection (Figs. 5a and 5b), and the simulated response of the
Z200 anomalies is generally larger than its counterpart at 500 hPa (Fig. 11b), which concurs with
the observational evidence. Besides, a similar low-level portion of the $A_{SJ}$ could also be simulated
(figure not shown). As a result, a strengthened $A_{SJ}$ is induced.
Figure 12 delineates the 850-hPa geopotential height (Z850) and UV850 responses to the specified
heating centered at (15.35°N, 109.69°E). Although there are some differences in spatial
distribution compared with the observations, the well-organized cyclonic anomaly to the west of
the heating center and the anticyclonic anomaly to the north can be properly simulated (Fig. 12).
Meanwhile, the $A_{SJ}$ and the coherent tropospheric subsidence over the BTH region and the
Northeast Asian anticyclonic anomaly were also simulated well (figure omitted), leading to the
amplified $A_{SJ}$ as well.
To sum up, from observational diagnoses and numerical simulations, we can conclude that there
are two pathways regarding how SSTAs impact the formation and maintenance of the $A_{SJ}$ and the
associated air subsidence. One pathway operates via a heating-induced large-scale teleconnection
pattern arising from SST warming in R1, and the other is connected to an anomalous local
meridional cell triggered by heating-reinforced ascending motion via local SST warming over R2.



## 5 Conclusions and discussion

Motivated by a lack of in-depth understanding with respect to the interannual variations of the $AHD_{BTH}$, in the present study we explored the related climate anomalies (localized meteorological parameters, and large-scale atmospheric and oceanic anomalies) tied to the $AHD_{BTH}$. We have substantiated that an above-normal $AHD_{BTH}$ is closely correlated with the simultaneous SST warming in two key regions (R1 over the North Atlantic subtropical sector, and R2 over the western North Pacific sector), and once the SON SST warming in R1 and R2 are both remarkably significant, their joint climate impacts can greatly enhance the likelihood of an above-normal $AHD_{BTH}$.

Potential mechanisms associated with an above-normal $AHD_{BTH}$ have been proposed through further investigations. Since the $A_{SJ}$ and the associated subsidence over the $A_{SJ}$ and the surrounding region can yield meteorological circumstances conducive to enhancing the likelihood of haze pollution in the BTH region, the issue of an above-normal $AHD_{BTH}$ can be reasonably transferred into uncovering how the SON $A_{SJ}$ and associated air subsidence are developed and sustained. We found that there are two possible pathways. First, SST warming in R1 can induce a downstream Rossby wave teleconnection, and the associated Rossby wave energy can propagate into the $A_{SJ}$ and its surrounding region through an arc-shaped trajectory, developing and strengthening the $A_{SJ}$ and the associated subsidence. The other pathway, however, operates through localized heating-reinforced ascending motion over R2, also resulting in subsidence over the BTH region and Northeast Asia via an anomalous local meridional cell.

AGCM simulations reinforced our hypothesis. With prescribed heating over the region to the north of R1, a quite similar teleconnection—starting from the North Atlantic subtropics—was excited. If we imposed an idealized heating over the adjacent R2, where the corresponding precipitation rate was the most significant and amplified, the concomitant significant low-level convergence around the heated areas was simulated, enhancing the SST warming in R2 and inducing the $A_{SJ}$-resembled circulation to the north and the subsidence over the BTH region and Northeast Asia. However, because the model we used is an intermediate anomaly AGCM, and the heating prescribed in the model is idealized, the simulated patterns were slightly spatially different to those observed. Although the model cannot reproduce the geopotential height and wind anomalies perfectly, it can nonetheless support the proposed mechanisms. As a summary, a schematic illustration (Fig. 13) of the occurrence of a higher $AHD_{BTH}$ is provided, which encapsulates the major characteristics of the two pathways of how SSTAs over R1 and R2 drive the $AHD_{BTH}$ respectively.

From the perspective of seasonal prediction, among all the previous individual months of boreal summer (June–July–August), the SON SST in R1 (R2) was most significantly correlated with the August SST in R1 (R2) on the interannual timescale, with a TCC of 0.35 (0.61) that exceeded the 95% (99%) confidence level. This suggests that, when the August SST over R1 (R2) is higher, the subsequent SON SST over R1 (R2) is more likely to become warmer. As such, the previous August SSTA over R1 (R2) could serve as a possible precursor for the seasonal prediction of the $AHD_{BTH}$.

In this study, we solely emphasize the potential impacts of SSTAs on the interannual variations of the $AHD_{BTH}$. It should be noted that other external forcings, such as the Arctic sea ice (e.g., Wang et al., 2015), Eurasian snowpack (e.g., Yin and Wang, 2018), thermal conditions on the Tibetan Plateau (e.g., Xu et al., 2016) and soil moisture (e.g., Yin and Wang, 2016b), may also exert profound impacts on haze pollution over China. Studying the mechanisms tied to these forcings may enhance the seasonal predicting skill for the $AHD_{BTH}$. This is an important topic deserving of





further exploration.

*Data availability.* The atmospheric data and land-surface data are available from the NCEP/NCAR data archive:
http://www.esrl.noaa.gov/psd/data/gridded/data.ncep.reanalysis.html (NCEP/NCAR, 2018). The SST data were downloaded from
https://www.esrl.noaa.gov/psd/data/gridded/data.noaa.ersst.v5.html (NOAA, 2018). The precipitation data were downloaded from
https://www.esrl.noaa.gov/psd/data/gridded/data.prec.html (NOAA, 2018). The monthly PBLH data are available on the ERA-Interim
website: http://www.ecmwf.int/en/research/climate-reanalysis/era-interim (ERA-Interim, 2018). The ground observations are from the
National Meteorological Information Center of China (http: //data.cma.cn/) (CMA, 2018).
*Competing interests.* The authors declare that they have no conflict of interest.
*Acknowledgements.* This work was supported by the National Natural Science Foundation of China (Grants 41605035, 41371222, and
41475086) and the Priority Academic Program Development (PAPD) of Jiangsu Higher Education Institutions. Zhiwei was supported by
the Natural Science Foundation of Jiangsu Province (No. BK20161604) and the Startup Foundation for Introducing Talent of NUIST (No.
2018r026).





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





## Figures Captions

**Figure 1.** Topographic map (shaded; m) for the BTH region and the locations of 20 meteorological sites (colored dots). The dots colored red (light red; magenta) represent significant positive temporal correlation coefficients at the 99% (95%; 90%) confidence level between the $AHD_{BTH}$ and AHD for every individual site on the interannual timescale.

**Figure 2.** Time series of the raw $AHD_{BTH}$ (black line; days), along with its interdecadal component (blue line; days) and interannual component (red line; days), for the period 1960–2017. The gray horizontal line delineates the average climate value of the raw $AHD_{BTH}$ during 1960–2017.

**Figure 3.** The climatological-mean (1960–2017) autumnal **(a)** Z500 (contours; gpm), UV850 (vectors; m s$^{-1}$) and total cloud (shaded; %), and **(b)** surface relative humidity (shaded; %) and surface air temperature (contours; ℃). The gray shaded area denotes the Tibetan Plateau, and the blue dashed box delineates the research domain of the BTH region. The letter A represents the center of anticyclonic circulation.

**Figure 4.** Regressed patterns of autumnal meteorological parameters onto the interannual component of the $AHD_{BTH}$, including **(a)** surface relative humidity (shaded; %), **(b)** surface air temperature (shaded; ℃), **(c)** surface wind speed (shaded; m s$^{-1}$), **(d)** SLP (shaded; hPa), **(e)** PBLH (shaded; m), and **(f)** 500-hPa omega (shaded; 10$^{-2}$ Pa s$^{-1}$). Regression coefficients that are significant at the 90% confidence level are stippled. The blue dashed box outlines the research domain of the BTH region.

**Figure 5.** Regressed anomalies of autumnal **(a)** 200-hPa geopotential height (Z200; shaded; gpm) and 200-hPa winds (UV200; vectors; m s$^{-1}$), **(b)** Z500 (shaded; gpm) and 500-hPa winds (UV500; vectors; m s$^{-1}$), **(c)** SST (shaded; ℃) and UV850 (vectors; m s$^{-1}$), **(d)** SLP (shaded; hPa) and surface winds (vectors; m s$^{-1}$), and **(e)** precipitation (shaded; mm day$^{-1}$), with respect to the interannual component of the $AHD_{BTH}$. Regression coefficients that are significant at the 95% (90%) confidence level are stippled (cross hatched). In panels **(a)** and **(b),** only the wind vectors with statistical significance above the 90% confidence level are shown. In panel **(c)**, the two red dashed rectangles, labelled R1 and R2, are the key regions where SSTAs are significantly correlated with the interannual component of the $AHD_{BTH}$; vectors with scales less than 0.05 m s$^{-1}$ are omitted. In panel **(d)**, vectors with scales less than 0.03 m s$^{-1}$ are omitted. The blue dashed box delineates the research domain of the BTH region. The letters A and C represent the centers of anticyclonic and cyclonic anomalies, respectively.

**Figure 6.** Time series of the normalized interannual component of the $AHD_{BTH}$ (black line), along with the simultaneous SST over R1 (red line) and R2 (blue line) for the period 1960–2017. The horizontal dashed lines denote 0.8 of the standard deviation. The numerals labelled at the bottom represent the correlation coefficients ($r$) between the $AHD_{BTH}$ and simultaneous SST over R1 and R2, separately. The upper and lower dots in the red line indicate the three highest and lowest years of SST over R1, respectively.

**Figure 7.** Regressed anomalies of autumnal UV850 (vectors; m s$^{-1}$) with respect to the simultaneous interannual component of the SST over R1. Green arrows represent the wind vectors with statistical significance above the 90% confidence level. The red dashed rectangle labelled R1 is the key region where SSTAs are significantly correlated with the interannual component of the $AHD_{BTH}$. The blue dashed box delineates the research domain of the BTH region. The gray shaded area denotes the Tibetan Plateau. The letters A and C represent the centers of anticyclonic and cyclonic anomalies, respectively.

**Figure 8.** The autumnal composite differences of **(a)** 200-hPa and **(b)** 500-hPa WAF (vectors; m$^2$ s$^{-2}$), geopotential height (contours; gpm), and relative vorticity (shaded; 10$^{-5}$ s$^{-1}$) between the three highest and three lowest years of simultaneous SST over R1 (highest minus lowest), as shown in Fig. 6. The red dashed rectangle labelled R1 is the key region where SSTAs are significantly correlated with the interannual component of the $AHD_{BTH}$. The blue dashed box delineates the research domain of the BTH region.

**Figure 9.** Regressed anomalies of autumnal **(a)** UV850 (vectors; m s$^{-1}$) and SST (shaded; ℃), and **(b)** precipitation (shaded; mm day$^{-1}$) with respect to the simultaneous interannual component of the SST over R2. In panel **(a),** green arrows represent the wind vectors with statistical significance above the 99% confidence level, and vectors with scales less than 0.05 m s$^{-1}$ are omitted. Regression coefficients that are significant at the 99% confidence level are cross hatched. The dashed red rectangle labelled R2 is the key region where SSTAs are significantly correlated with the interannual component of the $AHD_{BTH}$. The blue dashed box delineates the research domain of the BTH region. The gray shaded area denotes the Tibetan Plateau. The letter A (C) represents the center of anticyclonic (cyclonic) anomaly.

**Figure 10. (a)** Latitude–vertical section (112.5°–130°E) of the autumnal omega (shaded; 10$^{-2}$ Pa s$^{-1}$) and **(b)** longitude–vertical section (35°–42.5°N) of the autumnal air temperature (shaded; ℃) anomalies regressed onto the simultaneous interannual component of the SST over R2. Regression coefficients that are significant at the 90% confidence level are stippled. The thick blue horizontal bars superimposed onto the abscissa of panels **(a)** and **(b)** indicate the latitudes and longitudes of the BTH region, respectively.

**Figure 11.** The response of anomalous (a) Z200 (shaded; 10 gpm) and UV200 (vectors; m s$^{-1}$), and (b) Z500 (shaded; 10 gpm) and UV500 (vectors; m s$^{-1}$) in H_NAS. The red contours indicate the imposed idealized heating. The blue dashed box delineates the research domain of the BTH region. The letters A and C represent the centers of anticyclonic and cyclonic anomalies, respectively.

**Figure 12.** The response of Z850 (shaded; 10 gpm) and UV850 (vectors; m s$^{-1}$) in H_WNP. The magenta contours indicate the imposed idealized heating. The blue dashed box delineates the research domain of the BTH region. The gray shaded area denotes the Tibetan Plateau. The letter A (C) represents the center of anticyclonic (cyclonic) anomaly.

**Figure 13.** Schematic diagram encapsulating the SSTA-induced (warming in R1 and R2) physical mechanisms and pathways connected to above-normal $AHD_{BTH}$ years on the interannual timescale. Anomalous quasi-barotropic anticyclones (A) and cyclones (C) are indicated by blue and red elliptical cycles with arrows separately, denoting large-scale Rossby wave train triggered by the heating to the north of R1. Green arrows depict the key horizontal low-level (850-hPa) airflows. The red, azure and green arrows together exhibit the vertical overturning circulation tied to the SST warming in R2. The left-hand (right-hand) side of the cloud-resembled pattern with violet short dashed lines presents the significant anomalous precipitation induced by SSTAs over R1 (R2). The blue dashed box delineates the research domain of the BTH region.





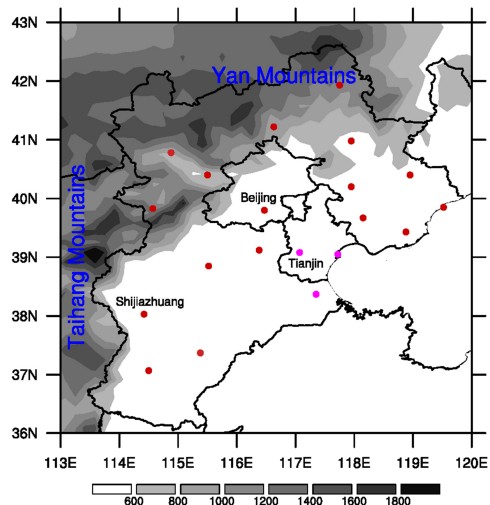


**Figure 1.** Topographic map (shaded; m) for the BTH region and the locations of 20 meteorological sites (colored dots). The dots colored red (light red; magenta) represent significant positive temporal correlation coefficients at the 99% (95%; 90%) confidence level between the $AHD_{BTH}$ and AHD for every individual site on the interannual timescale.





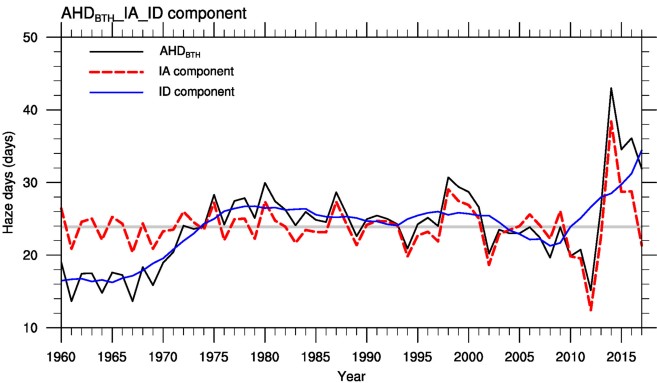

**735**

**Figure 2.** Time series of the raw AHD$_{BTH}$ (black line; days), along with its interdecadal component (blue line; days) and interannual component (red line; days), for the period 1960–2017. The gray horizontal line delineates the average climate value of the raw AHD$_{BTH}$ during 1960–2017.


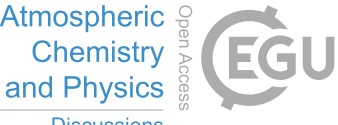



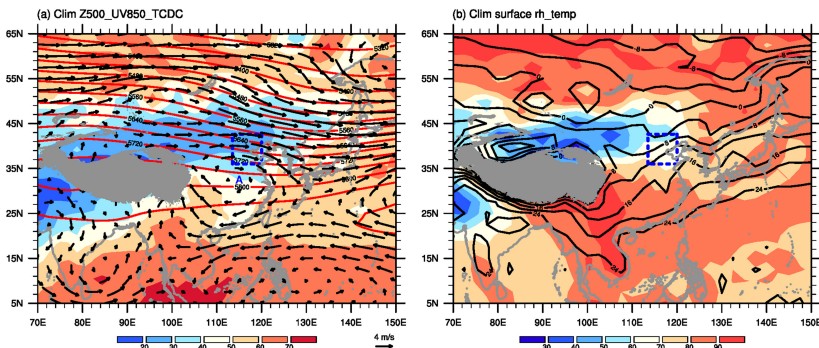

**Figure 3.** The climatological-mean (1960–2017) autumnal **(a)** Z500 (contours; gpm), UV850 (vectors; m s$^{-1}$) and total cloud (shaded; %), and **(b)** surface relative humidity (shaded; %) and surface air temperature (contours; ℃). The gray shaded area denotes the Tibetan Plateau, and the blue dashed box delineates the research domain of the BTH region. The letter A represents the center of anticyclonic circulation.





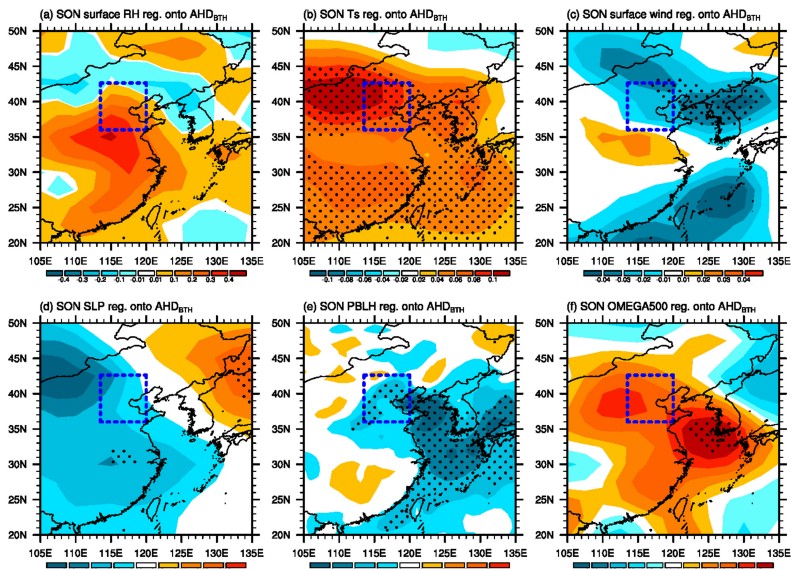


**Figure 4.** Regressed patterns of autumnal meteorological parameters onto the interannual component of the AHD$_{BTH}$, including **(a)** surface relative humidity (shaded; %), **(b)** surface air temperature (shaded; ℃), **(c)** surface wind speed (shaded; m s$^{-1}$), **(d)** SLP (shaded; hPa), **(e)** PBLH (shaded; m), and **(f)** 500-hPa omega (shaded; $10^{-2}$ Pa s$^{-1}$). Regression coefficients that are significant at the 90% confidence level are stippled. The blue dashed box outlines the research domain of the BTH region.




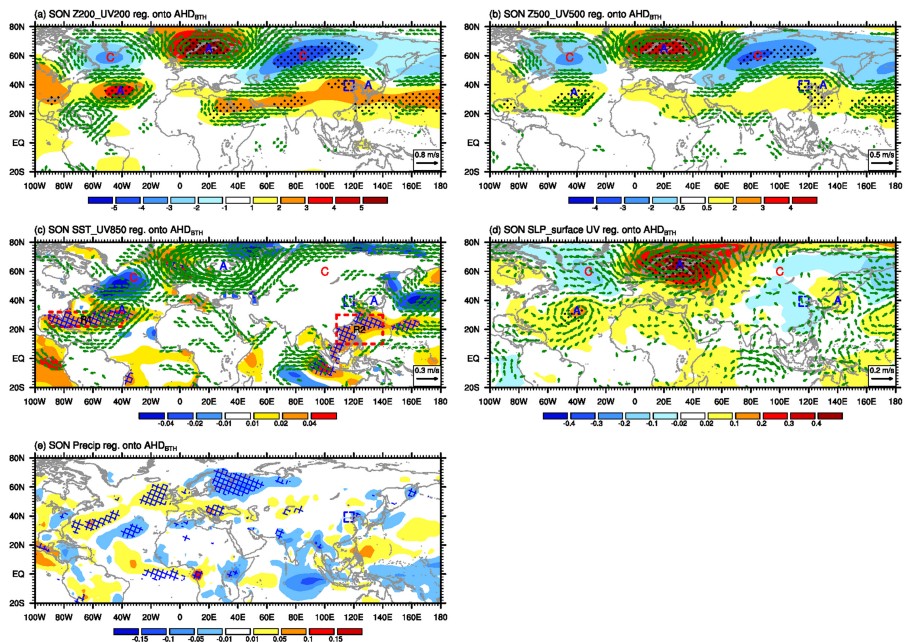


**Figure 5.** Regressed anomalies of autumnal **(a)** 200-hPa geopotential height (Z200; shaded; gpm) and 200-hPa winds (UV200; vectors; m s$^{-1}$), **(b)** Z500 (shaded; gpm) and 500-hPa winds (UV500; vectors; m s$^{-1}$), **(c)** SST (shaded; °C) and UV850 (vectors; m s$^{-1}$), **(d)** SLP (shaded; hPa) and surface winds (vectors; m s$^{-1}$), and **(e)** precipitation (shaded; mm day$^{-1}$), with respect to the interannual component of the AHD$_{BTH}$. Regression coefficients that are significant at the 95% (90%) confidence level are stippled (cross hatched). In panels **(a)** and **(b)**, only the wind vectors with statistical significance above the 90% confidence level are shown. In panel **(c)**, the two red dashed rectangles, labelled R1 and R2, are the key regions where SSTAs are significantly correlated with the interannual component of the AHD$_{BTH}$; vectors with scales less than 0.05 m s$^{-1}$ are omitted. In panel **(d)**, vectors with scales less than 0.03 m s$^{-1}$ are omitted. The blue dashed box delineates the research domain of the BTH region. The letters A and C represent the centers of anticyclonic and cyclonic anomalies, respectively.












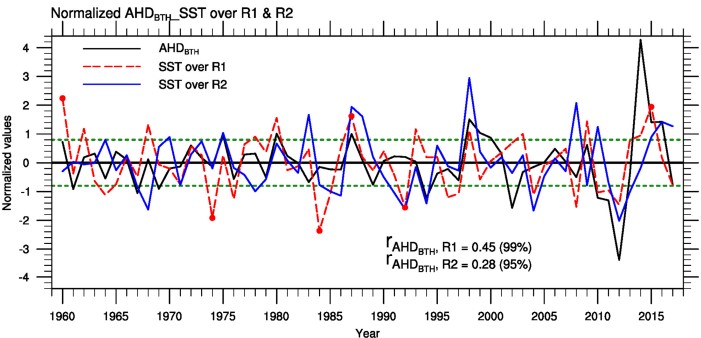


**Figure 6.** Time series of the normalized interannual component of the AHD$_{BTH}$ (black line), along with the simultaneous SST over R1
(red line) and R2 (blue line) for the period 1960–2017. The horizontal dashed lines denote 0.8 of the standard deviation. The numerals
labelled at the bottom represent the correlation coefficients ($r$) between the AHD$_{BTH}$ and simultaneous SST over R1 and R2, separately.
The upper and lower dots in the red line indicate the three highest and lowest years of SST over R1, respectively.






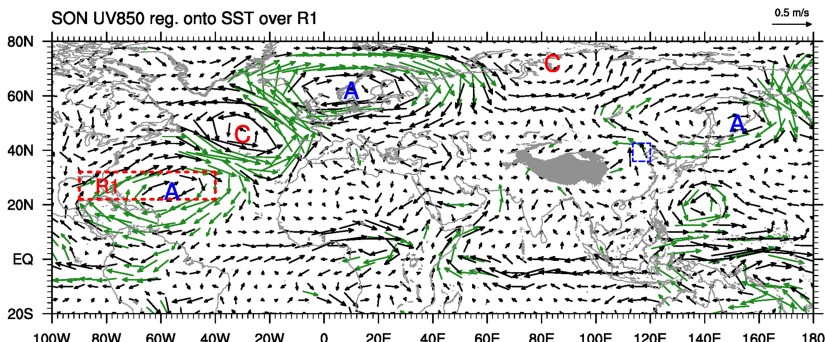

**Figure 7.** Regressed anomalies of autumnal UV850 (vectors; m s$^{-1}$) with respect to the simultaneous interannual component of the SST over R1. Green arrows represent the wind vectors with statistical significance above the 90% confidence level. The red dashed rectangle labelled R1 is the key region where SSTAs are significantly correlated with the interannual component of the AHD$_{BTH}$. The blue dashed box delineates the research domain of the BTH region. The gray shaded area denotes the Tibetan Plateau. The letters A and C represent the centers of anticyclonic and cyclonic anomalies, respectively.





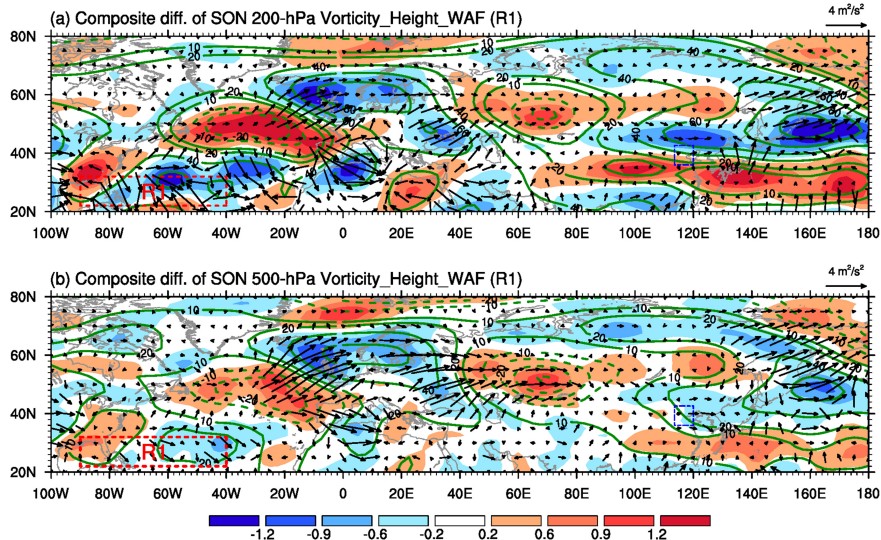

**Figure 8.** The autumnal composite differences of **(a)** 200-hPa and **(b)** 500-hPa WAF (vectors; $m^2 \ s^{-2}$), geopotential height (contours; gpm), and relative vorticity (shaded; $10^{-5} \ s^{-1}$) between the three highest and three lowest years of simultaneous SST over R1 (highest minus lowest), as shown in Fig. 6. The red dashed rectangle labelled R1 is the key region where SSTAs are significantly correlated with the interannual component of the $AHD_{BTH}$. The blue dashed box delineates the research domain of the BTH region.





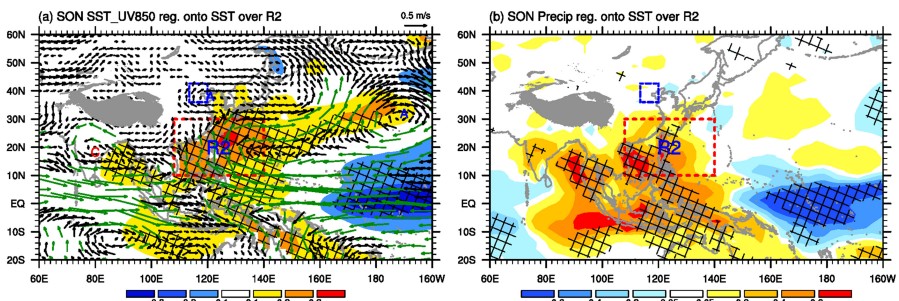


**Figure 9.** Regressed anomalies of autumnal **(a)** UV850 (vectors; m s$^{-1}$) and SST (shaded; ℃), and **(b)** precipitation (shaded; mm day$^{-1}$) with respect to the simultaneous interannual component of the SST over R2. In panel **(a)**, green arrows represent the wind vectors with statistical significance above the 99% confidence level, and vectors with scales less than 0.05 m s$^{-1}$ are omitted. Regression coefficients that are significant at the 99% confidence level are cross hatched. The dashed red rectangle labelled R2 is the key region where SSTAs are significantly correlated with the interannual component of the AHD$_{BTH}$. The blue dashed box delineates the research domain of the BTH region. The gray shaded area denotes the Tibetan Plateau. The letter A (C) represents the center of anticyclonic (cyclonic) anomaly.







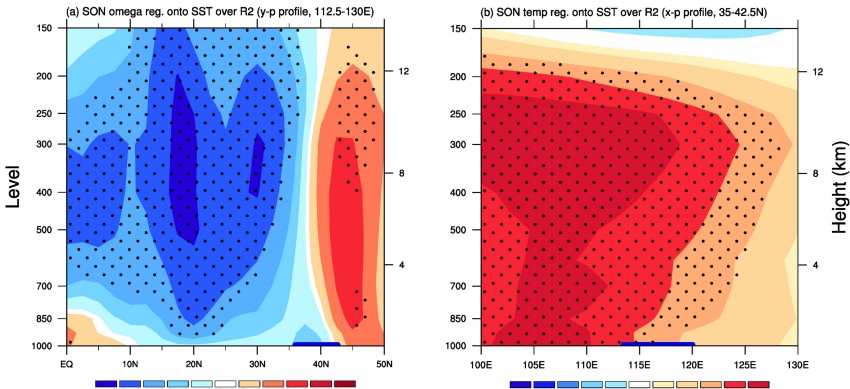

**Figure 10. (a)** Latitude–vertical section (112.5°–130°E) of the autumnal omega (shaded; $10^{-2}$ Pa s$^{-1}$) and **(b)** longitude–vertical section
(35°–42.5°N) of the autumnal air temperature (shaded; °C) anomalies regressed onto the simultaneous interannual component of the SST
over R2. Regression coefficients that are significant at the 90% confidence level are stippled. The thick blue horizontal bars superimposed
onto the abscissa of panels **(a)** and **(b)** indicate the latitudes and longitudes of the BTH region, respectively.

















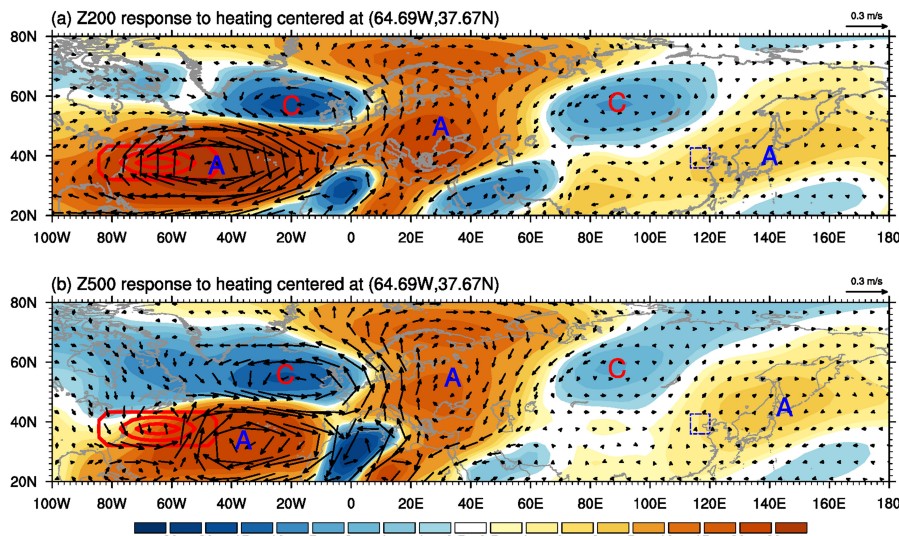


**Figure 11.** The response of anomalous (a) Z200 (shaded; 10 gpm) and UV200 (vectors; m s$^{-1}$), and (b) Z500 (shaded; 10 gpm) and
UV500 (vectors; m s$^{-1}$) in H_NAS. The red contours indicate the imposed idealized heating. The blue dashed box delineates the research
domain of the BTH region. The letters A and C represent the centers of anticyclonic and cyclonic anomalies, respectively.






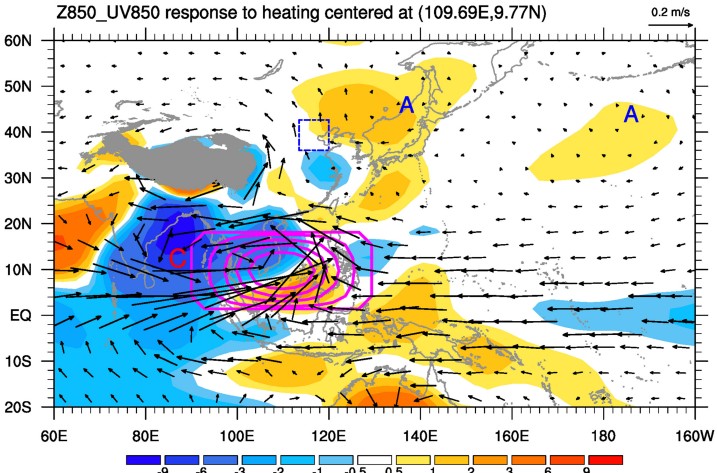


**Figure 12.** The response of Z850 (shaded; 10 gpm) and UV850 (vectors; m s$^{-1}$) in H_WNP. The magenta contours indicate the imposed idealized heating. The blue dashed box delineates the research domain of the BTH region. The gray shaded area denotes the Tibetan Plateau. The letter A (C) represents the center of anticyclonic (cyclonic) anomaly.






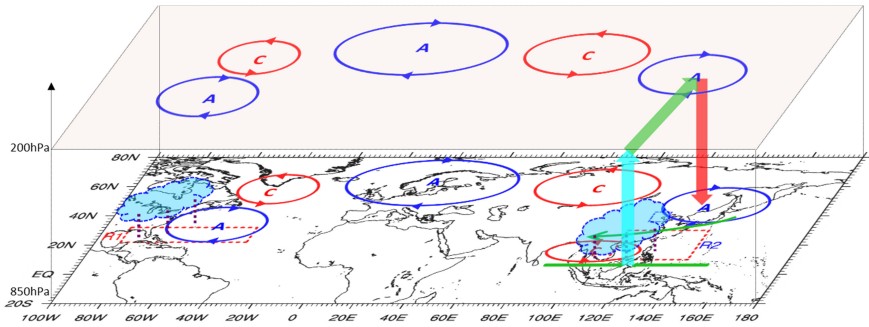


**Figure 13.** Schematic diagram encapsulating the SSTA-induced (warming in R1 and R2) physical mechanisms and pathways connected to above-normal AHD$_{BTH}$ years on the interannual timescale. Anomalous quasi-barotropic anticyclones (A) and cyclones (C) are indicated by blue and red elliptical cycles with arrows separately, denoting large-scale Rossby wave train triggered by the heating to the north of R1. Green arrows depict the key horizontal low-level (850-hPa) airflows. The red, azure and green arrows together exhibit the vertical overturning circulation tied to the SST warming in R2. The left-hand (right-hand) side of the cloud-resembled pattern with violet short dashed lines presents the significant anomalous precipitation induced by SSTAs over R1 (R2). The blue dashed box delineates the research domain of the BTH region.
