# Peer review of "Two pathways of how remote SST anomalies drive the interannual variability of autumnal haze days in the Beijing-Tianjin-Hebei region, China"

_Atmospheric Chemistry and Physics, 2018_

## Referee Comment (RC1) · Anonymous Referee #2 · 21 Jan 2019

This paper provides a new possible signal source in the North Atlantic subtropical sector (R1) and the western North Pacific sector (R2) for autumnal haze days (AHD) in the Beijing-Tianjin-Hebei region (BTH region) via the tele-connection mode. The effect sequence of the warm phase of these two oceanic sources on the AHD in BTH is basically reasonable, leading to depressed planetary boundary layer and subsidence of the atmosphere. These changing meteorological conditions are the favorable background for the higher AHD in BTH. The methodology used in this paper is correct (i.e. Rossby wave train). The findings obtained by this paper may be useful to make the seasonal outlook of the air pollution condition in autumn. The specified corrections: (1) For the SST of the North Atlantic, why only the middle oceanic region is selected? The

representative signal source of the AMO should be the triple-pole SST pattern, with high-latitude and the tropical poles being more important. (2) From Figure 2, one see the rapid increase of AHD in BTH. However, other studies have shown that the rapid increase in AHD started from the mid-ninety of 20 century or early in this century. Please compare the difference between them and explain why. (3) For Figure 6, please indicate the significance level for the correlation coefficient. (4) The anticyclonic circulation over Northeast China-Okhotsk Sea at 850 hPa is a critical system. Please check if it only takes place in autumn (or/and winter)? Whether or not it already exist in summer? (5) From Figure 10(a), the descending motion seems to be out of BTH region. Please explain it. (6) The warm R2 should be associated with the El Nino event. Therefore, according the EAP pattern, it could be "- + -" meridional circulation pattern. Please attention to this point and further properly modify the position A in Figure 13.

---

## Referee Comment (RC2) · Anonymous Referee #1 · 21 Jan 2019

The impact of external thermal forcing induced atmospheric circulation changes on air quality is an important issue in atmospheric environment study. Focusing this scientific issue, this manuscript presented an interesting finding on two pathways of thermal forcing sources in the North Atlantic region (R1) and the western North Pacific region (R2) drive the interannual variations of autumnal haze pollution in the air pollution region of North China via the tele-connection analysis and AGCM simulation, which could improve our understanding and prediction on air quality change in China, Asia and the Northern Hemisphere. This manuscript falls within the scope of ACP. I suggest the minor revisions before it is published as follows:

1) Please add the discussions on the tele-connection pattern from the R1 region to North China in connection with North Atlantic Oscillation, and the R2 region in association with Western Pacific Warm Pool.

2) Please modify the lines 20-21: the joint impacts can greatly enhance the likelihood of a higher $AHD_{BTH}$ Observational and simulation evidence suggests that SST anomalies can affect the variation……

3) Lines 191 and 193, please add "surface air" before "temperature".

4) Please add the box outlines the research domain of the BTH region in Fig. 7.

---

## Author Comment (AC1) · 25 Jan 2019

*Reply to Referee #1*

*General comments:*

*The impact of external thermal forcing induced atmospheric circulation changes onair quality is an important issue in atmospheric environment study. Focusing this scientific issue, this manuscript presented an interesting finding on two pathways of thermal forcing sources in the North Atlantic region (R1) and the western North Pacific region (R2) drive the interannual variations of autumnal haze pollution in the air pollution region of North China via the tele-connection analysis and AGCM simulation,which could improve our understanding and prediction on air quality change in China, Asia and the Northern Hemisphere. This manuscript falls within the scope of ACP. I suggest the minor revisions before it is published as follows:*

**Reply:** Thank you for your positive comments. We have revised the manuscript based on your comments/suggestions. Below is our point-by-point reply to these comments/suggestions (italic is for original comments and non-italic is our replies).

*Specific comments:*

*1)Please add the discussions on the tele-connection pattern from the R1 region to North China in connection with North Atlantic Oscillation, and the R2 region in association with Western Pacific Warm Pool.*

**Reply:** Thanks for your comment. We have added two relevant references and corresponding discussions. Please see Lines 208-211 and Lines 241-242 in the revised manuscript for the discussions.

(Lines 208-211) "Intriguingly, from the surface projection of the above quasi-barotropic teleconnection pattern, a positive phase of North Atlantic Oscillation-like pattern (Hurrell and Deser, 2009) can be discerned which is part of the teleconnection."

(Lines 241-242) "with its large portion belonging to the region of Western Pacific Warm Pool (You et al., 2018)."

**Reference:**
Hurrell, J. W., and Deser, C.: North Atlantic climate variability: The role of the North Atlantic Oscillation, J Marine Syst, 78, 28-41, 10.1016/j.jmarsys.2008.11.026, 2009.
You, Y. C., Cheng, X. G., Zhao, T. L., Xu, X. D., Gong, S. L., Zhang, X. Y., Zheng, Y., Che, H. Z., Yu, C.,Chang, J. C., Ma, G. X., and Wu, M.: Variations of haze pollution in China modulated by thermal forcing of the Western Pacific Warm Pool, Atmosphere, 9, 314, 10.3390/atmos9080314, 2018.

**2)**_Please modify the lines 20-21: the joint impacts can greatly enhance the likelihood of a higher AHD$_{BTH}$ Observational and simulation evidence suggests that SST anomaliescan affect the variation……_

**Reply:** Thanks for your comment.The modification was done. Please see Lines 19-22 in the revised manuscript.

Lines 19-22: "When the autumnal SST warming in R1 and R2 are both significant, the likelihood of a higher AHD$_{BTH}$ is greatly enhanced. Observational and simulation evidence demonstrated how SST anomalies over R1 and R2 influence variation of AHD$_{BTH}$ via two different pathways."

**3)**_Lines 191 and 193, please add "surface air" before "temperature"._

**Reply:** Thanks for your suggestion. We have added "surface air" before "temperature". Please see Lines 194 and 196 in the revised manuscript.

**4)**_Please add the box outlines the research domain of the BTH region in Fig. 7._

**Reply:** Thanks for your constructive suggestion. We have added the blue dashed box outlining the research domain of the BTH region. Please see Fig. 7 in the revised manuscript.

---

## Author Comment (AC2) · 25 Jan 2019

*General comments:*

*This paper provides a new possible signal source in the North Atlantic subtropical sector (R1) and the western North Pacific sector (R2) for autumnal haze days (AHD) inthe Beijing-Tianjin-Hebei region (BTH region) via the tele-connection mode. The effect sequence of the warm phase of these two oceanic sources on the AHD in BTH is basically reasonable, leading to depressed planetary boundary layer and subsidence of the atmosphere. These changing meteorological conditions are the favorable background for the higher AHD in BTH. The methodology used in this paper is correct (i.e.Rossby wave train). The findings obtained by this paper may be useful to make the seasonal outlook of the air pollution condition in autumn.*

**Reply:** Thank you for your positive comments. We have revised the manuscript based on your comments/suggestions. Below is our point-by-point reply to these comments/suggestions (italic is for original comments and non-italic is our replies).

*Specified corrections:*

*(1) For the SST of the North Atlantic, why only the middle oceanic region is selected? The representative signal source of the AMO should be the triple-pole SST pattern, with high-latitude and the tropical poles being more important.*

**Reply:** Thanks for your comment. The AMO is known as the SST in the North Atlantic varying on the basin scale and at period of around 65–80 years. Since our study concentrated on the interannual variability, the AMO is not relevant to our research target.

We chose subtropical North Atlantic region ($22\degree$–$32\degree$N, $90\degree$–$40\degree$W) as the key SSTA region for the following two reasons. Firstly, the subtropical North Atlantic SSTA is the only region over North Atlantic that highly correlated with the $AHD_{BTH}$ on interannual timescale. Although the regression SSTA pattern over North Atlantic looks like a tri-pole SST pattern (NAT SST pattern for short) which has profound impacts on Eurasian climate (e.g., Chen and Wu, 2017), the relationship between $AHD_{BTH}$ and simultaneous NAT SST pattern is insignificant. The correlation coefficient between $AHD_{BTH}$ and NAT SST triple-pole index (Deser and Michael, 1997) is only 0.17. Therefore, we chose the middle oceanic region of North Atlantic as the key region for $AHD_{BTH.}$

Secondly, the positive correlated SSTA over that region can induce positive rainfall anomaly (diabatic heating). Therefore, the SSTA should play an active role in local air-sea interaction and in turn influence the large-scale circulation through inducing teleconnection.

To sum up, we chose subtropical North Atlantic region (22 °–32 °N, 90 °–40 °W) as the key driving region from both statistical diagnosis and physical basis.

**Reference:**

Chen, S.F. and Wu, R.G., 2017. Interdecadal changes in the relationship between interannual variationsof spring North Atlantic SST and Eurasian surface air temperature. Journal of Climate, 30(10): 3771-3787.

Deser, C. and Michael S.T., 1997: Atmosphere-ocean interaction on weekly timescales in the North Atlantic and Pacific. Journal of Climate, 10(3): 393-408.

*(2) From Figure 2, one can see the rapid increase of AHD in BTH. However, other studies have shown that the rapid increase in AHD started from the mid-ninety of 20 century or early in this century. Please compare the difference between them and explain why.*

**Reply:** Thanks for your constructive comments. After checking many related literatures (e.g., **Figure 6b** in Fu and Dan, 2014), we found that the annual total haze days indeed showed the rapid increasein the BTH region since the early 21st century, but it is not materialized for the case of autumn season.

For the autumn season, from the previous studies, we found that the rapid increase in AHD over North China did not start from the mid-ninety of 20 century or early in this century (R-**Figure 1**). Our results are quite consistent with the previous studies.

[Figure]

**R-Figure 1**. Time series of AHD in North China. (left) Adapted from ***Chen and Wang*** (**2015**). (right) Adapted from ***Ding and Liu*** (**2014**). The red dashed line is 9-point smooth curve.

**Reference:**

Chen, H. P., and Wang, H. J.: Haze Days in North China and the associated atmospheric circulations based on daily visibility data from 1960 to 2012, J Geophys Res Atmos, 120, 5895-5909, 10.1002/2015JD023225, 2015.

Ding, Y. H., and Liu, Y. J.: Analysis of long-term variations of fog and haze in China in recent 50 years and their relations with atmospheric humidity, Sci China Earth Sci, 57, 36-46, 10.1007/s11430-013-4792-1, 2014.

Fu, C. B., and Dan, L.: Spatiotemporal characteristics of haze days under heavy pollution over central and eastern China during 1960–2010, Climatic and Environmental Research (in Chinese), 19 (2), 219-226, 2014.

*(3) For Figure 6, please indicate the significance level for the correlationcoefficient.*

**Reply:** Thanks for your suggestion. We have added the significance level for the correlation coefficient as suggested. Please see Figure 6 in the revised manuscript.

*(4) The anticyclonic circulation over Northeast China-Okhotsk Sea at 850 hPa is a critical system. Please check if it only takes place in autumn (or/and winter)? Whether or not it already exists in summer?*

**Reply:** Thanks for your good comment and suggestion. As you indicated, the 850-hPa anomalous anticyclonic circulation over Northeast China-Okhotsk Sea is indeed a critical system that having significant impacts upon the interannual variability of $AHD_{BTH}$. In addition to autumn season, this system also takes place in winter (e.g., **Figure 4b** in Yin et al., 2017; **Figure 3a** in Zhong et al., 2018), which greatly influences the interannual changes of wintertime haze pollution.

However, this anticyclonic circulation does not exist in the prior summer based on our correlation analysis (figures omitted). Therefore, we could infer that this anticyclonic circulation anomaly over Northeast Asia is only occurred in the simultaneous autumn and winter.

**Reference:**

Yin, Z. C., Wang, H. J., and Chen, H. P.: Understanding severe winter haze events in the North China Plain in 2014: roles of climate anomalies, AtmosChemPhys, 17, 1641-1651, 10.5194/acp-17-1641-2017, 2017.

Zhong, W. G., Yin, Z. C., Wang, H. J., and: The Relationship between the Anticyclonic Anomalies in Northeast Asia and Severe Haze in the Beijing–Tianjin–Hebei Region, AtmosChem Phys, Discuss., 10.5194/acp-2018-782, in review, 2018.

*(5) From Figure 10(a), the descending motion seems to be out of BTH region. Please explain it.*

**Reply:** Thanks for your comment. Except for a small portion of upward motion over the southern BTH region, most of the BTH region is indeed dominated by air subsidence from mid-to-upper troposphere (700-300 hPa in Figure 10a).

The significant descending motion is tied to strong ascending motion over south of the BTH region. Because the ascending motion is so strong, it makes the descending motion farther to the north. Nevertheless, this subsidence could enhance Northeast Asian anticyclonic anomaly.

*(6) The warm R2 should be associated with the El Nino event. Therefore, according the EAP pattern, it could be "- + -" meridional circulation pattern. Please attention to this point and further properly modify the position A in Figure 13.*

**Reply:** Thanks for your insightful suggestion. The El Niño event related to EAP pattern (or PJ pattern) mainly appears in boreal summer. The present study focuses on the autumn season. We think Northeast Asian anticyclonic anomaly is not part of the EAP pattern, but rather a joint result of the mid-latitude Rossby wave train emanating from Atlantic to East Asia, and an anomalous local meridional circulation forced by western North Pacific SSTA.